Preferred temperature and thermal breadth of birds wintering in peninsular Spain: the limited effect of temperature on species distribution

Carrascal Luis M. lmcarrascal@mncn.csic.es 1
Villén-Pérez Sara 1 2
Palomino David 3
1 Department of Biogeography and Global Change, Museo Nacional de Ciencias Naturales—CSIC , Madrid , Spain
2 Departamento de Ecologia, Universidade Federal de Goiás , Goiânia , Brazil
3 Wildlife Consultor , Madrid , Spain
Wink Michael
Electronic publication date: 2016 Jul 19
Publication date: 2016
Volume: 4
Electronic Location ID: e2156
Received 2016 Mar 17; Accepted 2016 May 31
Copyright: ©2016 Carrascal et al.
Copyright year: 2016
Copyright holder: Carrascal et al.
License: This is an open access article distributed under the terms of the Creative Commons Attribution License, which permits unrestricted use, distribution, reproduction and adaptation in any medium and for any purpose provided that it is properly attributed. For attribution, the original author(s), title, publication source (PeerJ) and either DOI or URL of the article must be cited.
License URL: https://creativecommons.org/licenses/by/4.0/

Keywords: Quantile regression, Preferred temperature, Bird abundance, Species distribution, Thermal breadth, Winter

Funding: Spanish Ministry of Education and Science BES-2009-029386 Nacional Council of Scientific and Technological Development of Brazil 300274/2015-7 During the development of this work, Sara Villén-Pérez was supported by a ‘Formacion de Personal Investigador’ predoctoral fellowship (BES-2009-029386) of the Spanish Ministry of Education and Science and by a ‘Young Talent Attraction’ fellowship (300274/2015-7) of the Nacional Council of Scientific and Technological Development of Brazil associated to project 401408/2014-0. David Palomino was contracted by SEO/BirdLife to organize, design, analyze and develop the Winter Spanish Bird Atlas. The funders had no role in study design, data collection and analysis, decision to publish, or preparation of the manuscript.

==============================
Background. The availability of environmental energy, as measured by temperature, is expected to limit the abundance and distribution of endotherms wintering at temperate latitudes. A prediction of this hypothesis is that birds should attain their highest abundances in warmer areas. However, there may be a spatial mismatch between species preferred habitats and species preferred temperatures, so some species might end-up wintering in sub-optimal thermal environments.

Methods. We model the influence of minimum winter temperature on the relative abundance of 106 terrestrial bird species wintering in peninsular Spain, at 10 ×10 km2 resolution, using 95%-quantile regressions. We analyze general trends across species on the shape of the response curves, the environmental preferred temperature (at which the species abundance is maximized), the mean temperature in the area of distribution and the thermal breadth (area under the abundance-temperature curve).

Results. Temperature explains a low proportion of variation in abundance. The most significant effect is on limiting the maximum potential abundance of species. Considering this upper-limit response, there is a large interspecific variability on the thermal preferences and specialization of species. Overall, there is a preponderance of positive relationships between species abundance and temperature; on average, species attain their maximum abundances in areas 1.9 °C warmer than the average temperature available in peninsular Spain. The mean temperature in the area of distribution is lower than the thermal preferences of the species.

Discussion. Many species prefer the warmest areas to overwinter, which suggests that temperature imposes important restrictions to birds wintering in the Iberian Peninsula. However, one third of species overwinter in locations colder than their thermal preferences, probably reflecting the interaction between habitat and thermal requirements. There is a high inter-specific variation in the versatility of species using the available thermal space, and the limited effect of temperature highlights the role of other environmental factors determining species abundance.

Introduction

The distribution of overwintering animals is assumed to be strongly influenced by environmental energy availability, notably in regions with a marked year-round seasonality such as temperate ecosystems. Peninsular Spain is an important destination for many avian populations of the southwestern Palearctic during the winter (Moreau, 1972), when there are massive migrations of northern populations towards circum-Mediterranean countries. Even though conditions are milder than in the north, it is an energy-limiting period in which food resources are scarce and/or difficult to locate and the low temperatures impose a high metabolic cost to maintain a constant body temperature for homeothermic animals (Calder & King, 1974). Specifically, average winter temperature in peninsular Spain (7.2 °C) is well below the lower critical temperature for a broad variety of bird species (ca. 18–22 °C, Calder & King, 1974; Kendeigh, Dol’nik & Gavrilov, 1977). Thus, winter survival depends primarily on a positive energy balance, obtaining enough food for self-maintenance and reducing metabolic costs of thermoregulation (e.g., Newton, 1998). In this ecological scenario, species abundances are expected to reach their maxima in warmer areas, through the direct effects of reduced thermoregulation costs and reduced mortality by hypothermia, or indirectly via the improvement of the winter foraging environment (e.g., Gosler, 1996; Doherty & Grubb, 2002; Rogers & Reed, 2003; Robinson, Baillie & Crick, 2007; Cresswell, Clark & Macleod, 2009; Carrascal, Villén-Pérez & Seoane, 2012). However, the availability of “optimal environments,” combining preferred temperatures together with other habitat or trophic requirements, may be restricted. As a consequence, the environments in which the individuals of a species end up wintering may show a mean temperature different from the thermal preference of the species (at which the maximum abundances are attained). On the other hand, although general patterns are expected in relation to thermal optima, there might be notable interspecific differences on the thermal tolerance of species. Thermal breadth of species may define how individuals utilize the thermal space and ultimately the geographical area occupied by species (Slatyer, Hirst & Sexton, 2013).

Moreover, while the abundance of species may be limited by temperature at some points, it might be further limited by other environmental factors related to species-specific habitat or trophic preferences at other locations (Herrando et al., 2011; SEO/BirdLife, 2012; Howard et al., 2015). Thus, the correlation between species abundance and temperature may often display a solid distribution as that shown in Fig. 1A. The upper limit of this distribution is defined by locations in which temperature is the factor actually limiting abundance, while the points below this limit correspond to locations in which other environmental factors are limiting abundance further than temperature. The upper limit of these distributions would represent the maximum potential abundance of species attainable at each environmental temperature, which in theory is independent of other environmental factors (Cade & Noon, 2003; Fig. 1A).

Figure 1 Representation of environmental preferred temperature (TPREF), mean temperature (TMEAN) and thermal breadth (TBREATH) of an example specie.

(A) Abundance of Columba palumbus in relation to minimum winter temperature along 1,689 UTM cells, and fitting curves for quantile regression models (from top to bottom: models on 95th, 75th, 50th and 25th percentiles). Relative abundance is the number of 15 min transects over 60 in which the species is detected at each UTM 10 × 10 km2 cell. (B) TPREF, TMEAN and TBREATH of Columba palumbus. Environmental preferred temperature (TPREF) is the temperature at which the maximum abundance of the species is predicted by the quantile regression model for percentile 95th in (A). Mean temperature (TMEAN) is the mean winter minimum temperature in those UTM cells where the species was present, weighed by the relative abundance of the species at each cell. Thermal breadth (TBREATH) is the standardized area under the curve of quantile regression model for percentile 95th in (A), from −2 to 10 °C (shaded area in B).

The first goal of this study is to test the hypothesis that minimum winter temperature—as a surrogate of environmental energy availability—limits the maximum potential abundance of terrestrial birds wintering in the Iberian Peninsula, so that warmer environments will have the potential to maintain a higher number of individuals. Specifically, we test two predictions of this hypothesis: (1) that the maximum abundance of species will correlate positively with minimum winter temperature, (2) that, on average, species will prefer temperatures above the mean environmental temperature available in the region. The second goal is to test the hypothesis that as a consequence of species being limited by other factors (e.g., habitat preferences, food availability), mean temperatures at which species are found (TMEAN) do not coincide with the preferred temperature at which the species abundance is maximized (TPREF). Finally, we analyze the interspecific variation on the level of specialization to use the available thermal space (i.e., the thermal breadth of species, TBREADTH; Fig. 1B).

We modeled the influence of minimum winter temperature on the abundance of 103 species of terrestrial birds wintering in the Iberian Peninsula using quadratic 95%-quantile regression models (Fig. 1A). To analyze macroecological patterns in the abundance-temperature relationship across species, we use the standardized regression coefficients and two parameters derived from quantile regression models: the ‘environmental preferred temperature’ of species, calculated as the temperature at which its abundance is maximal within the thermal span of the study region (TPREF), and the ‘thermal breadth’ of species, calculated as the area under the response curve relative to the maximum abundance of the species (TBREADTH Fig. 1B). These measures are able to detect higher inter-specific variability in both thermal preferences and thermal breadth of species than other classical approaches (Villén-Pérez & Carrascal, 2015).

Materials and Methods

Bird abundance and temperature data

We obtained field data from the national-scale project conducted by SEO/BirdLife (2012) for the first Spanish Atlas of Wintering Birds. See Fig. 2B for the geographical location of the study area within the Western Palearctic. We calculated the ‘relative abundance’ of each species at 1,689 UTM 10 × 10 km2 cells, with very good sampling cover, as the frequency of occurrence in sixty 15-min transects sampled throughout three consecutive winters (see a summary of the methods in the Spanish Atlas of Wintering Birds in Text S1). We selected 103 bird species for the analyses, excluding nocturnal and aquatic birds, species that were detected in less than 50 UTM 10 × 10 km2 cells, and those that were rare or very difficult to detect (i.e., those with a maximum recorded frequency of occurrence lower than 0.05, or three 15-min transects per 60 transects censused).

Figure 2 Minimum winter temperature (°C) and relative abundance of three example bird species in peninsular Spain, and relationship between these variables.

(A) Minimum winter temperatures in the study area (peninsular Spain). (B) Location of the study area (black) within the western Palearctic (dark grey). (C, E, G) Winter relative abundance of three sample species (Saxicola rubicola, Erithacus rubecula and Turdus viscivorus, respectively), at 1,689 UTM 10 × 10 km2 cells within the study area, sampled in three consecutive winters (2008–2011). Relative abundance is the frequency of occurrence in sixty 15-min linear transects carried out in each UTM cell. (D, F, H) Relationship between the relative abundance of these species and minimum winter temperature, as modeled by quadratic 95%-quantile regression models.

The Meteorological Spanish agency (www.aemet.es) provided updated GIS temperature data covering the whole Iberian Peninsula, from which we calculated the average minimum winter temperature of each UTM cell as the daily averages during the period of study (mid-November to mid-February 2007–2010). Average minimum winter temperature during the study period in the 1,689 UTM cells considered was highly correlated with average winter temperature (r = 0.978) and average maximum winter temperature (r = 0.909). The average minimum winter temperature was selected as a measurement of the thermal state of the environment more probably constraining bird distribution and abundance, considering its functional meaning related to maximum thermoregulatory costs. Moreover, minimum temperatures usually occur at night, which is the most constraining period for diurnal species during winter, considering the long duration of nights, foraging inactivity and the lack of heat production from locomotion (Carrascal, Seoane & Villén-Pérez, 2012).

Data analyses

We analyzed the influence of minimum winter temperature (T) on the relative abundance (A) of each species using quantile regression models at percentiles 50th, 75th, 90th and 95th (i.e., τ = 0.50, 0.75, 0.90, 0.95; see Cade & Noon, 2003 and references therein, Fig. 1A). To account for non-linear effects of temperature, we defined the linear and quadratic terms of the relationship (A = a + bT + cT2). We standardized the original temperature variable T, and its squared term T2, to mean = 0 and sd = 1 prior to data analyses, in order to obtain standardized regression coefficients, also called beta regression coefficients. Under this standardization, the magnitude of beta coefficients allows to compare the relative contribution of each independent variable in the prediction of the response variable because predictors are on the same measurement scale. We estimated pseudo-R2 for each quantile regression model as a goodness-of-fit measure (Koenker & Machado, 1999), analogous but not exactly homologous to R2 in least square models (i.e., how well the quantile regression represents the variability observed in the response variable; a higher pseudo-R2 indicates a better fit). In addition, we calculated the increase in AIC of these models with respect to the null model as a measure of the likelihood of the model (Burnham & Anderson, 2002).

We obtained the environmental preferred temperature (TPREF) by solving the equation dA⋅dT−1 = 0 in quadratic 95%-quantile regression models. Then, we calculated the mean temperature in the area of distribution (TMEAN) as the weighted average of the winter minimum temperature of the UTM cells where the species were present, using the relative occurrence of species in the sixty 15-min transects. Finally, we obtained the thermal breadth of species (TBREADTH) by integrating A⋅dT in quadratic 95%-quantile regression models between −2 and 10 °C, standardizing the maximal abundances of all species to 1 (Fig. 1B); this index ranges from 0 to 1.

We carried out all data analyses with R 1.8-10 (R Core Team, 2015), using packages quantreg version 5.21 (Koenker, 2015) and Hmisc version 3.17-3 (Harrell, 2015). See Fig. 2 for three example species, showing different patterns of relationship between relative abundance and average winter minimum temperature. The script we employed in analyses is found in Data S1.

Results

Relationships between average winter minimum temperature and abundance show a wedge-shaped point cloud in all species, with values varying from zero to an upper limit of abundance (Figs. 1 and 2). Minimum winter temperature explains an average of 2.7% (se = 0.534), 4.8% (0.773), 7.1% (0.924) and 8.7% (0.964) of species abundances using quantile regression models at percentiles 50%, 75%, 90% and 95%, respectively (see pseudo-R2 for quantile regression models of all species in Table S1). Only in ten out of 103 species, 95%-quantile regression models attain figures of pseudo-R2 higher than 25%, while it is lower than 5% in 50 species. There is a significant increase in pseudo-R2 from the percentile 50% to 95% (repeated measure ANOVA testing for the linear contrast of increase from 50% to 95%: F1,102 = 64.49, p ≪ 0.001).

In 93 out of 103 species, the 95%-quantile regression models including the linear and quadratic terms of temperature attain AIC figures that are 13.82 units lower than those AIC figures obtained for 95%-quantile null regression models (i.e., the temperature models are 1,000 times better in explaining the variation in relative abundance of the species than the null models; 1,000 = exp\nolimits [ − 0.5∗13.816]; Burnham & Anderson, 2002). In other ten species the ΔAIC is higher than −6 (see Table S1).

Standardized lineal regression coefficients b in the 93 species with “significant” 95%-quantile regression models are on average positive, and significantly different from zero (t test = 4.994, df = 92, p ≪ 0.001; Table 1, Table S1). Standardized quadratic regression coefficients c show predominantly negative values, on average significantly different from zero (across-species comparison: t test = − 3.144, df = 92, p = 0.002), defining a hump-shaped relationship between temperature and the relative abundance of bird species. Linear terms b have larger absolute values than the quadratic terms c (average of absolute figures of b and c: 4.97 and 3.05, respectively; paired t test: t = 5.51, df = 92, p ≪ 0.001). Therefore, the linear increase of relative abundance with winter temperature is, on average, positive and more important than the curvilinear pattern defining maxima.

Table 1 Parameters of the response of species abundance to winter temperature.

Figures are mean, standard deviation and range of parameters derived from 95%-quantile regression models describing the influence of minimum winter temperature on abundance of bird species wintering in peninsular Spain, sampled at 1,689 UTM 10 × 10 km2 cells in three consecutive winters (2008–2011). Sample size is 93 species when considering only significant models with a reduction in AIC figures (ΔAIC) lower than −13.82 units, and 103 species when significance of models is not relevant and therefore all species are considered. Detailed data for all species are shown in Table S1.

	Mean	sd	Range	n	
Standardized linear coefficient, b	3.22	6.22	−9.38/18.94	93	
Standardized curvilinear coefficient, c	−1.43	4.39	−18.12/7.54	93	
Environmental preferred temperature (°C), TPREF	4.36	4.72	−2/10	93	
Mean temperature on distribution areas (°C), TMEAN	2.75	1.10	−0.2/5.5	103	
Thermal breadth, TBREADTH	0.64	0.20	0.26/1	103	
Notes.

b, c, linear and quadratic regression coefficients obtained from 95%-quantile regression models on the effect of minimum winter temperature on the relative abundance of species; TPREF, winter minimum temperature at which the relative abundance of the species is maximized; TMEAN, mean of average winter minimum temperature in those UTM cells where the species were present, weighed by the relative abundance of the specie at the cell; TBREADTH, area under the curve defined by the second order polynomial equation that relates the relative abundance of species to the temperature using the coefficients of the 95%-quantile regression models. n, number of species considered.

The average TPREF is 4.36 °C for 93 species with “significant” 95%-quantile regression models (range: −2 °C to 10 °C; Table 1, Table S1), and this average is significantly higher than the average minimum environmental temperature available during winter in peninsular Spain (2.55 °C; t-test: t = 3.70, df = 92, p < 0.001). TPREF is lower than 0 °C in 24 out of 93 species (i.e., preferences for colder areas; e.g., Dryocopus martius, Cinclus cinclus, Turdus pilaris, Serinus citrinella, Fringilla montifringilla, Emberiza cia), while it is higher than 5 °C in 40 species (i.e., preferences for warmer areas; e.g., Elanus caeruleus, Upupa epops, Alcedo atthis, Burhinus oedicnemus, Ptynoprogne rupestris, Phylloscopus collybita, Troglodytes troglodytes, Cisticola juncidis, Sylvia undata; see Table S1).

Mean temperature in the area of distribution (TMEAN) is 2.75 °C for the 103 studied species (range: −0.20 °C to 5.51 °C; Table 1, Table S1). In 77 out of 103 species TMEAN is significantly different from the average minimum temperature available in winter in peninsular Spain (2.55 °C; significant t-tests after the sequential Bonferroni correction), with 35 bird species whose distribution correspond to colder conditions than average, and 42 species inhabiting warmer areas than average. Other 26 species do not show any clear, significant, preference for warmer or colder areas in peninsular Spain.

TPREF and TMEAN are highly correlated (r = 0.856, n = 93, p ≪ 0.001; Fig. 3), although TPREF has, on average, higher values than TMEAN (paired t-test: t = 3.83, df = 92, p ≪ 0.001). In fact, there are 32 species with TPREF > 8 °C that show a TMEAN 4.5–8.3 °C colder. Conversely, there are 21 species with TPREF = − 2°C and a TMEAN 1.8–4.1 °C higher.

Figure 3 Relationship between TPREF and TMEAN for 93 bird species wintering in peninsular Spain.

The graph shows 93 species for which the 95%-quantile regression models including the linear and quadratic terms of temperature attained AIC figures that were 13.82 units lower than those AIC figures obtained for 95%-quantile null regression models. Solid line represents the linear regression between TPREF and TMEAN.

TBREADTH is on average 0.64 for all studied species (range: 0.26–1.00, n = 103 species; see Table 1 and Table S1). It is low (i.e., thermal specialists, <0.33) in species such as Dryocopus martius, Oenanthe leucura, Turdus pilaris, Remiz pendulinus, Serinus citrinella, and high (i.e., thermal generalists, >0.90) in species such as Accipiter nisus, Turdus merula, Parus major, Corvus monedula, Carduelis cannabina, Fringilla coelebs (see Table S1). In those ten species in which 95%-quantile regression models are “non-significant” the average thermal breadth is 0.93, both facts indicating the independence of the distribution of these species with respect to temperature.

Discussion

The maximum abundance of birds wintering in the Iberian Peninsula is influenced by average winter minimum temperature in 90% of the studied species. As a general trend, the relative abundance of species increases with temperature and, on average, species reach their maximum abundances at temperatures 1.9 °C warmer than the average winter minimum temperature available in the study region. Nevertheless, temperature alone can only explain a small proportion of variation in maximum abundance, and the shapes of point clouds suggest that other environmental factors that were not included in temperature models, such as food availability or habitat structure, may be more relevant to explain species abundance. The relative importance of temperature depends on the species: the thermal breadth of the studied species varies from 0.26 to 1.00, reflecting a broad spectrum from thermal specialists to thermal generalists (Table S1; Moussus et al., 2011; Barnagaud et al., 2012).

In a winter scenario with temperatures well below the thermoneutral zone, warmer environments may significantly reduce bird metabolic costs and improve the foraging environment, overall reducing winter mortality rates (Calder & King, 1974; Kendeigh, Dol’nik & Gavrilov, 1977; Root, 1988; Canterbury, 2002; Meehan, Jetz & Brown, 2004; Cresswell, Clark & Macleod, 2009; Zuckerberg et al., 2011). Nevertheless our results show that the relationships between bird abundance and temperature are variable and idiosyncratic (see also Reif et al., 2010; La Sorte & Jetz, 2012; Fraixedas, Lehikoinen & Linden, 2015). For instance, 33% of species show statistical significant preferences for environments colder than average conditions (see TPREF in Table S1). Contrary to the general positive relationships between winter temperature and bird abundance, which are easy to explain according to thermoregulatory costs and food accessibility, these negative relationships are hardly explainable using metabolic arguments for endotherms in wintertime. There might exist other important aspects of bird natural history, such as specialized food preferences or selection for particular habitats with a restricted spatial distribution that are responsible for the emergence of those negative relationships between temperature and animal abundance. This may be the case of resident species with restricted habitat preferences, such as for example very mature and extensive forests (e.g., Dryocopus martius), mountain streams (Cinclus cinclus), or alpine rock outcrops (Prunella collaris; Herrando et al., 2011), and species with a very specialized diet such as the fruits of the Spanish juniper tree (Juniperus thurifera) that grows in highlands of continental cold climate (Turdus torquatus, T. pilaris, T, viscivorus; Tellería, Carrascal & Santos, 2014), or seeds provided by pine species with small cones in montane coniferous forests (Serinus citrinella; Borras et al., 2010). If these habitats and food types are unequivocally linked with areas of cold climate, then the negative relationship of those species with temperature may be the casual consequence of functional responses to habitats and food resources (Barnagaud et al., 2012).

Our results suggest that temperature has little importance in limiting winter bird distribution (average pseudo-R2 of 8.7% for the 95%-quantile regression models, and only ten species with pseudo-R2 > 25%). The steady increase of pseudo-R2 from the median (50% quantile) to the maximum response (95% quantile) shows that the influence of temperature on bird distribution is more clearly revealed at the upper edge of the wedge-shaped pattern of covariation abundance—temperature, where the limiting effect of temperature surpasses that of other factors affecting bird abundance (see Fig. 1A). The detected meager influence of temperature on the spatial variation of winter bird abundance is consistent with results obtained in other European areas (Reif et al., 2010; Dalby et al., 2013; Fraixedas, Lehikoinen & Linden, 2015), suggesting that other factors such as food and habitat availability are more important governing winter bird distribution in this region of the southwestern Palearctic (see also Carrascal, Villén-Pérez & Seoane, 2012; Carrascal, Seoane & Villén-Pérez, 2012 for the competing effects of food, vegetation and temperature on the winter abundance of small passerines at smaller spatial scales in the Iberian Peninsula).

Quantile regression is a method of analyzing the unequal variation in a variable of interest along a set of predictor of variables when there are multiple rates of change (or slopes) from the minimum to the maximum response (Cade, Terrell & Schroeder, 1999; Cade & Noon, 2003). This approach allows the identification of limiting factors, paying more attention to the slopes near the maximum response (e.g., maximum abundance attained at each temperature), which provides a thorough picture of the patterns of covariation between the animal abundance and temperature. Thus, the estimation of the response of a high quantile of population density to a measured predictor variable is generally considered to be a better estimate of the effect of that variable as a limiting factor than the estimate of the response to the mean calculated with least squares. This is because other unmeasured variables may be the active limiting constraint in the dependent variable of interest, through their correlations with the measured predictor (Borsuk, 2008). For example, if a UTM cell has a winter temperature that approaches the thermal preference of a species but lacks the habitat with the vegetation structure characteristics and food availability that configure the spatial-trophic niche of the species, the species should be probably very scarce in that UTM cell (e.g., Sylvia melanocephala may be scarce in a warm cell with minimum winter temperature 9 °C but lacking Mediterranean maquis with high abundance of ripe fruits). That sample unit will occupy a low position in the wedge-shaped pattern depicted by Fig. 1A. Therefore, estimating the upper edge of the wedge-shaped pattern of covariation abundance—temperature allows for the identification of the limiting effect of temperature on bird abundance, disregarding the probable interactions between temperature and other limiting predictors (measured or unmeasured). This is a sound concern, as the influence of temperature on bird distribution and abundance is probably mediated through surrogate effects of spatial variables, habitat preferences or resource availability (see Aragón et al., 2010 for direct and indirect effects of climatic and non-climatic factors on distribution of ectothermic and endothermic vertebrates in the Iberian Peninsula). For example, Repasky (1991) found little evidence to support that the northern distributions of North American wintering birds are governed principally by temperature, suggesting that temperature probably plays a role through interactions with biotic factors such as food, habitat structure and competition. The importance of these interactions on bird abundance distribution, is clearly shown by the differences between the mean winter minimum temperature in those UTM cells where the species were present (TMEAN) and the preferred temperature (TPREF) derived from quadratic 95%-quantile regression models (Fig. 3). Although both parameters are highly correlated, most individuals end-up overwintering in locations that are colder than the species thermal preferences, which may reflect a limitation of sites combining thermal and other environmental optima. For instance, insectivorous small passerines, such as Cettia cetti, Sylvia undata, Motacilla alba or Saxicola rubicola, occupy areas of peninsular Spain that are ca. 6 °C colder than their preferred temperatures (see Table S1), probably because they lack their preferred habitats in those warm areas. Considering this evidence, the low use of quantile regressions in the study of animal distribution patterns in relation to climate is highly surprising, a fact that may be a constraint in ecologists’ ability to analyze the influence of climatic variables for elucidating the underlying patterns (Austin, 2007; Vaz et al., 2008).

The general preference for warmer environments that we found suggest that winters will be less restrictive for most birds wintering in the Iberian peninsula under future climate warming scenarios (IPCC, 2007; Brunet et al., 2009; Stocker et al., 2013), though the impact of changes will depend on species-specific thermal preferences and plasticity (Khaliq et al., 2014; Pearce-Higgins et al., 2015; Gaüzère, Jiguet & Devictor, 2015). Zuckerberg et al. (2011) showed for birds wintering in North America that average minimum temperature is an important factor limiting bird distributions, and that local within-winter extinction probabilities are lower, and colonization probabilities higher, at warmer sites, supporting the role of climate-mediated range shifts. Climate warming may be especially beneficial for those species with narrow thermal breadths that prefer higher winter temperatures and that mainly rely on arthropods and fruits as winter food (e.g., Upupa epops, Ptynoprogne rupestris, Troglodytes troglodytes, Luscinia svecica, Cisticola juncidis, Phylloscopus collybita, Sylvia melanocephala, S. atricapilla). In the same vein, Tellería, Fernández-López & Fandos (2016) found that according to temperature increase projections for 2050–2070, two insectivorous passerines wintering in the Western Mediterranean basin (Anthus pratensis and Phylloscopus collybita) will broaden their distribution ranges into the cold highland expanses typical of the western Mediterranean. ‘Space-for-time’ substitution when forecasting temporal trends from spatial climatic gradients must be taken with care (La Sorte et al., 2009), as well as the potential interaction of climate and habitat changes on species responses (Chamberlain et al., 2013). Nevertheless, these forecasts on bird distributions are supported by the analyses of recent avian populations and winter minimum temperature changes in North America, where shifting winter climate has provided an opportunity for smaller, southerly distributed species to colonize new regions (Prince & Zuckerberg, 2015).

Conclusions

This study highlights the high interspecific variability on the response to temperature and thermal tolerance. Bird species wintering in peninsular Spain range from the coldest to the warmest thermal preferences and from thermal specialists to generalists. Nevertheless, the general trend is to select the warmest areas, so that abundance of most species increases with temperature and is predicted to reach its maximum at temperatures higher than the average available temperature in the study region. Even though species attain their maximum abundances at warm areas, a large proportion of the area of distribution of the studied species is in colder locations, probably reflecting a limitation of sites combining thermal optima with other environmental preferences of a species. Our results suggest that minimum winter temperature functions as a physiological constraint on the maximum potential abundance attainable by a species at a certain location, though other environmental factors may be more relevant than temperature defining the biogeographical patterns of species.

Supplemental Information

Text S1 Description of the Spanish Bird Atlas of Winter Birds

Click here for additional data file.

Data S1 Script for R environment employed in analyses and dataset including four example species

Click here for additional data file.

Table S1 Parameters defining the relationship between relative abundance and minimum winter temperature utilized in the study and obtained for 103 terrestrial bird species wintering in the Iberian Peninsula, corresponding to winter censuses on 1,689 10 × 10km2 UTM (years 2008–2011)

Click here for additional data file.

Juan Carlos del Moral (SEO/BirdLife) provided the raw data from the Winter Spanish Bird Atlas. We also thank Jorge M Lobo for helpful comments on a first draft of the paper and C Jasinski for improving the English of the manuscript. This paper is a contribution to projects CGL2008-02211/BOS and CGL2011-28177/BOS of the Spanish Ministry of Education and Science.

Additional Information and Declarations

Competing Interests

Author Contributions

Data Availability

The authors declare there are no competing interests.

Luis M. Carrascal conceived and designed the experiments, performed the experiments, analyzed the data, contributed reagents/materials/analysis tools, wrote the paper, prepared figures and/or tables, reviewed drafts of the paper.

Sara Villén-Pérez conceived and designed the experiments, performed the experiments, analyzed the data, wrote the paper, prepared figures and/or tables, reviewed drafts of the paper.

David Palomino contributed reagents/materials/analysis tools, prepared figures and/or tables, organized, designed, analyzed and developed the Winter Spanish Bird Atlas.

The following information was supplied regarding data availability:

The raw data has been supplied as Data S1.

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
