# Peer review of "Preferred temperature and thermal breadth of birds wintering in peninsular Spain: the limited effect of temperature on species distribution"

_PeerJ, doi:10.7717/peerj.2156_

## Round 0.1 · original submission · Minor Revisions

· Academic Editor

Minor Revisions

Dear authors
Your ms has now been reviewed and the reviewers suggest a number of corrections. Please follow their advice when you resubmit your ms.

Please discuss that temperatures alone can only explain some degree of the distribution. Consider the availability of food which correlates often with food plants. For them not the mean temperature counts but the extremes.

Kind regards

Michael Wink
Academic editor

Reviewer 1 ·

Basic reporting

The paper is clearly structured and written.

Experimental design

The methods employed were well presented and justified. The focus is on quantile regression which is a good way to analyze the data although one must take care to interpret the results properly.

Validity of the findings

The analyses are carried out and described in sufficient detail. I only would like to point out the difficulty in assessing the coefficients of the quadratic terms in the regression. Since they scale the quadratic, one would expect them to be lower than the linear term what makes a direct comparison of the absolute values a bit ambiguous (lines 161-162). At least figures 1 and 2d show, I would say, a clear dominance of the quadratic term

Additional comments

The interpretation of the results stresses that other factors than temperature may govern species’ abundances. I agree with that and wonder whether this is the most important conclusion to draw from the results. The relationships between temperature and abundance depicted in the figures clearly indicate that temperature and/or closely associated factors are related to the variance in the abundance data rather than a measure of location such as mean or median. In fact, the median curve in Figure 1 is rather flat. This increased variance also implies higher maximum values. One should also note the frequent occurrence of zero valued abundances. Could one interpret this pattern by assuming that high and low temperatures (and/or related variables) are acting as constraints while intermediate temperatures do not and other factors determine abundance and thus result in these high variance domains.
I am not sure whether I understood the very last sentence of the Conclusion and what it is intended to express. I have the feeling that it somewhat contradicts the preceding one.
Lines 47-49: How relevant is this? Consider that the average lowest temperatures in S-Sweden are between 7 and 11°C during the breeding season.
Line 157: “on average”: does this say much?
Several sections indicate the low importance of temperature as such (lines 208-209, 248-249, 263-265), however, I got the admittedly subjective impression that the authors feel that this is embarrassing.

Reviewer 2 ·

Basic reporting

No Comments

Experimental design

No Comments

Validity of the findings

No Comments

Additional comments

In general, I consider the MS an interesting contribution. Please try to avoid too general statements such as before the Conclusions.

Relatively few papers were cited. Imran Khaliq recently wrote about thermal niches of and the impact of climate change on bird and mammal species.

Use past tense for the methods.

Why are there so many species with preferred temperature at -2 and 10 °C, respectively (Fig. 3)?

Some minor comments:
19, 48: use SI units separated from the preceding number by a blank, e.g. km²
42: better use destination instead of target
135-136: cite R and each R package separately using the version numbers
210: that are responsible OR that are the responsible ones
313: of a species
454: Fig. 2b shows Europe, not the W Palearctic

---

## Round 0.2 · accepted · Accept

· Academic Editor

Accept

Dear authors

Good news- your revision is accepted and your paper will be published soon. Thanks for submitting your work to PeerJ

Regards
Michael Wink
Academic editor